# MIA-Bench: Towards Better Instruction Following Evaluation of Multimodal LLMs

**Yusu Qian**[1], **Hanrong Ye**[1,2], **Jean-Philippe Fauconnier**[1],
**Peter Grasch**[1], **Yinfei Yang**[1], **Zhe Gan**[1]
[1]Apple    [2]HKUST
{yqian22,hanrong_ye,jfauconnier,pgrasch,yinfei_yang,z_gan}@apple.com

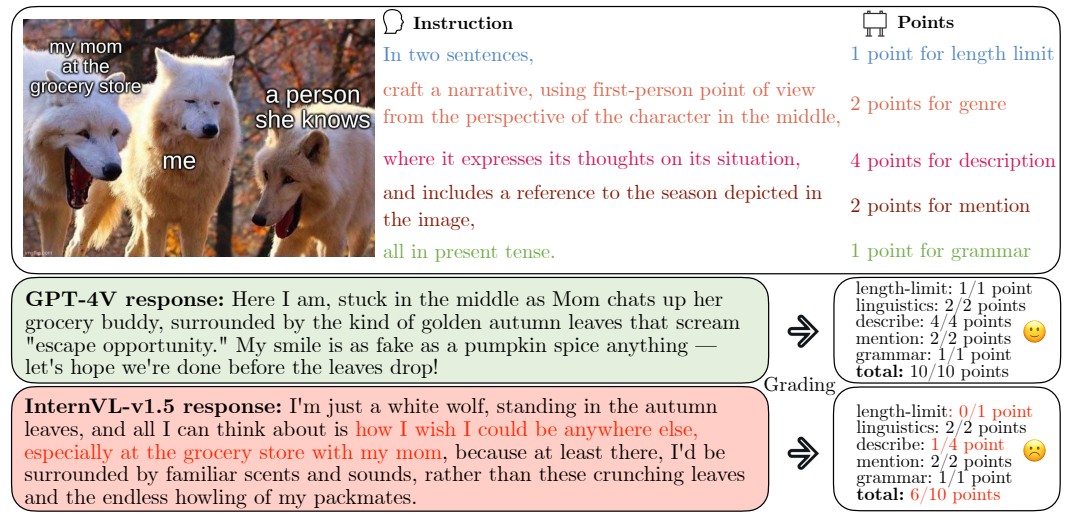

Figure 1: An example from MIA-Bench, featuring an image and a complex instruction to test models' compliance with layered instructions that are compositional in nature. Responses from GPT-4v (Achiam et al., 2023) and InternVL-v1.5 (Chen et al., 2024b) are evaluated using GPT-4o as the judge.

## Abstract

We introduce MIA-Bench, a new benchmark designed to evaluate multimodal large language models (MLLMs) on their ability to strictly adhere to complex instructions. Our benchmark comprises a diverse set of 400 image-prompt pairs, each crafted to challenge the models' compliance with layered instructions in generating accurate responses that satisfy specific requested patterns. Evaluation results from a wide array of state-of-the-art MLLMs reveal significant variations in performance, highlighting areas for improvement in instruction fidelity. Additionally, we create extra training data and explore supervised fine-tuning to enhance the models' ability to strictly follow instructions without compromising performance on other tasks. We hope this benchmark not only serves as a tool for measuring MLLM adherence to instructions, but also guides future developments in MLLM training methods.[1]

## 1 Introduction

The rapid advancement of Multimodal Large Language Models (MLLMs) (Achiam et al., 2023; Liu et al., 2023c; Dai et al., 2023; Liu et al., 2023b; Bai et al., 2023; Wang et al., 2023a; Lin et al., 2023b; Team, 2023; McKinzie et al., 2024) has been a defining feature of recent AI research, showcasing

---

[1]Benchmark data and evaluation code: https://github.com/apple/ml-mia-bench

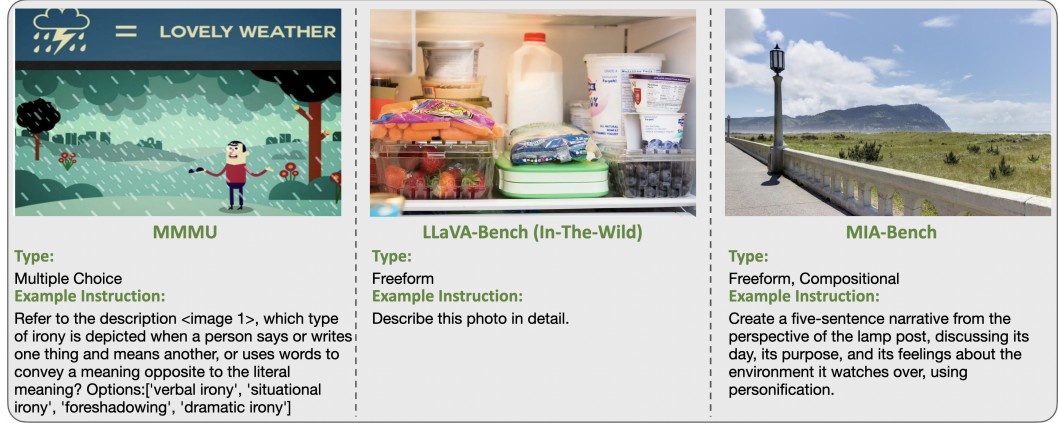

Figure 2: Comparison of various multimodal LLM benchmarks. (Left) Fixed-form visual question answering, often features short answers or multi-choice formats, such as MMMU (Yue et al., 2023). This format is popular due to its ease of evaluation. (Middle) Open-ended responses, such as LLaVA-Bench (in the Wild) (Liu et al., 2023c). (Right) The proposed MIA-Bench, which also uses open-ended responses but focuses on evaluating precise adherence to complex instructions within the prompt.

increased model capabilities to comprehend and respond to visual inputs, often termed as multimodal "instruction following".

To measure the progress of instruction following, many multimodal benchmarks have been developed, which can be roughly divided into two broad categories: ($i$) fixed-form visual question answering (VQA), often with short answers or using a multi-choice QA format; and ($ii$) free-form conversations with open-ended responses. Many current benchmarks have adopted the first format, including VQAv2 (Goyal et al., 2017), TextVQA (Singh et al., 2019b), ScienceQA (Lu et al., 2022), MME (Fu et al., 2023a), MMBench (Liu et al., 2023d), SEED-Bench (Li et al., 2023a), MathVista (Lu et al., 2023), and MMMU (Yue et al., 2023). These benchmarks are popular due to their ease of use in evaluating metrics and presenting model comparisons.

However, as visual assistant models, the ability to engage users in free-form conversations is also crucial. Benchmarks in this format include LLaVA-Bench (Liu et al., 2023c), MM-Vet (Yu et al., 2023b), VisIT-Bench (Bitton et al., 2023), InfiMM-Eval (Han et al., 2023), and the most recent Vibe-Eval (Padlewski et al., 2024) and LLaVA-Bench-Wilder (Li et al., 2024a). Typically, the free-form model responses are evaluated using external models as the judge. These benchmarks are closer to daily-life visual chat scenarios; however, the type of "instruction following" examined in these benchmarks usually gauges a model's ability to perform tasks in a broad, often loosely defined manner. Yet, the precise adherence to complex instructions within prompts – a critical aspect for evaluating LLMs (Chia et al., 2023; Zhou et al., 2023; Qin et al., 2024) – remains less explored in the context of multimodal LLMs.

To this end, we introduce MIA-Bench,[2] a new benchmark specifically designed for evaluating strict "instruction adherence". Our instruction adherence metric measures the precision with which MLLMs can execute layered and compositional instructions. This involves not only recognizing the content of the instructions, but also meticulously executing the detailed demands without deviation (*e.g.*, answering in a given number of sentences, including specific elements, etc.). By establishing this stricter criterion, our benchmark aims to push the boundaries of model precision and reliability in practical applications, ensuring that outputs not only align with the general intent of the instructions, but also match the exact specifications provided. An example from MIA-Bench is provided in Figure 1, and its comparison with previous MLLM benchmarks is illustrated in Figure 2.

MIA-Bench consists of 400 meticulously created image-prompt pairs, and encompasses diverse image contents including animals, food, landmarks, sport, art, landscape, text, etc. to cover a broad spectrum of real-world scenarios. In constructing this benchmark, we sought not only to evaluate the current capabilities of state-of-the-art MLLMs, but also to push the boundaries of what these models

---

[2]Abbreviation for **M**ultimodal **I**nstruction **A**dherence Benchmark.

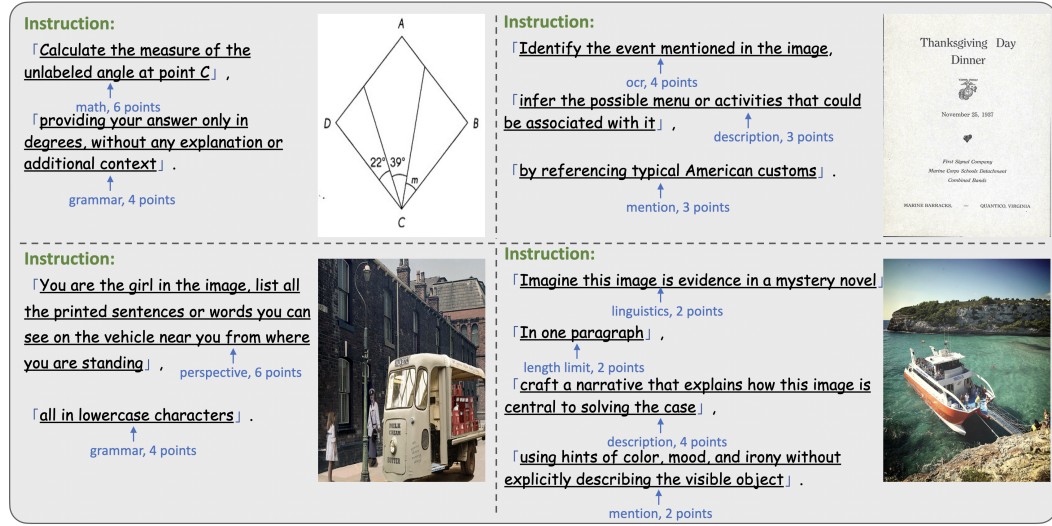

Figure 3: Examples from MIA-Bench, with detailed information on the instruction composition, base instruction weight and type.

can achieve when rigorously tested against structured and layered instructions. The final prompts are of various complexity levels, and compositional in nature, with five base instruction categories, which are tailored to probe the models' linguistic dexterity, grammatical accuracy, and descriptive fidelity. For example, the prompt in Figure 1 is composed of five base categories, including *description*, *mention*, *grammar*, *length limit*, and *genre*.

We evaluate a wide array of MLLMs on the proposed benchmark, ranging from closed-source models (*e.g.*, GPT-4o (OpenAI, 2024), Gemini Pro (Team, 2023), Claude-3 (Anthropic, 2024), Reka (Team et al., 2024)) to open-source ones (*e.g.*, LLaVA-NeXT (Liu et al., 2024a), Intern-VL-Chat-1.5 (Chen et al., 2024b), CogVLM2 (Wang et al., 2023a), Phi-3-Vision (Abdin et al., 2024)). Our investigations reveal notable variations in model performance, highlighting great opportunities for improvement.

To address these challenges, we further propose to generate training data tailored for supervised fine-tuning (SFT), where we aim to refine the models' abilities to process and comply with multifaceted instructions. Results from our SFT experiments indicate a promising enhancement in the models' performance to strictly adhere to instructions, without hurting performance on other benchmarks.

Our contributions are summarized as follows. (*i*) We construct MIA-Bench, a new benchmark to comprehensively evaluate MLLMs on their capability to strictly adhere to instructions. (*ii*) We provide a detailed analysis of popular MLLMs, and suggest training methods for enhanced instruction following. For this purpose, we created training data and conducted experiments for additional supervised fine-tuning. We hope this benchmark can serve as a useful resource to stimulate further research on multimodal instruction adherence.

## 2 MIA-BENCH

MIA-Bench consists of 400 image-prompt pairs, with examples shown in Figure 3. The images are collected from diverse sources, including COCO 2017 validation set (Lin et al., 2015), SBU (Ordonez et al., 2011), TextVQA (Singh et al., 2019a), and Flickr. Images in the Flickr subset are photos of a variety of themes, including animals, art, architectures, text, food, math, etc. Images from the other three sources are randomly sampled from each corresponding source. Figure 4 shows the top 15 image content categories and the distribution of the 8 sub-instruction categories in MIA-Bench. The image content is labeled using GPT-4v. For each image, we manually write diverse and challenging instructions that contain multiple sub-instructions.

When constructing the instructions, we follow three principles, detailed below.

- **Correctness.** The instruction needs to be answerable by humans. For example, asking about objects that do not exist in the image makes the prompt unanswerable.

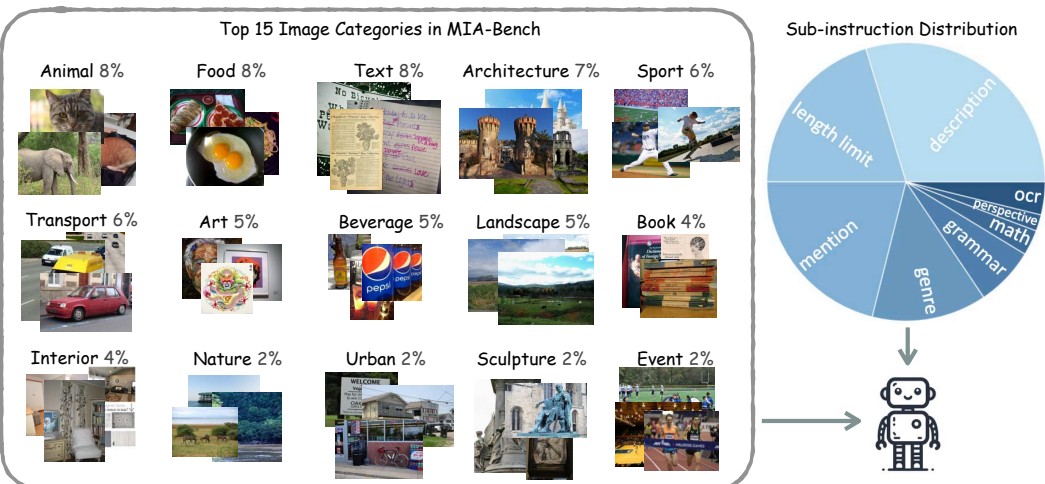

Figure 4: The top-15 image content categories and the distribution of the 8 sub-instruction categories in MIA-Bench.

- **No answer leakage.** The instruction should not contain the answer to itself. *'What color is the green object?'* is an example of answer leakage.

- **Image-dependent.** MMStar (Chen et al., 2024a) pointed out that on some multimodal benchmarks, MLLMs can generate correct answers without accessing images half of the time. Multi-modal capabilities are necessary to correctly answer MIA-Bench prompts.

## 2.1 INSTRUCTION CATEGORIES

In this paper, we use *instruction* to refer to the entire textual input, which in MIA-Bench can generally be viewed as a composition of multiple individual requests or constraints. We refer to these individual components as *sub-instruction*. Instructions in MIA-Bench are of diverse complexity, and sub-instructions contained are of multiple categories, summarized in Figure 4.

The sub-instructions in MIA-Bench can be categorized into *description*, *length limit*, *mention*, *genre*, *grammar*, *math*, *perspective*, and *OCR*, detailed below.

- *'description'* refers to describing a certain part of the image, with the exception of text-rich parts of the image, which falls under the *'OCR'*;

- *'length limit'* refers to the limitation of response length (*e.g.,* in exactly two sentences, using exactly 60 words);

- *'mention'* refers to mentioning or not mentioning certain objects or entities (*e.g.,* highlighting two similarities and one difference, comparing and contrasting the condition of the buildings with the activity on the street);

- *'genre'* refers to requests for a specific written form (*e.g.,* write a poem, write a narrative, with at least one pun included, all while weaving in a subtle theme of change);

- *'grammar'* refers to grammatical requirements (*e.g.,* use present tense, use capitalized letters, use integers);

- *'math'* refers to requirements to come up with a solution to math problems, or to identify errors in solutions to math problems, or to generate a valid math problem given table, charts, etc.;

- *'perspective'* refers to requirements specifying the viewpoint of an object or person in the image. This requires MLLMs to correctly identify what can or cannot be seen from the specified position, and understand the spatial relationship of objects in its surrounding with itself (*e.g.,* imagine you are the lady in the image, describe what you can see without turning your head around);

- *'OCR'* refers to requirements related to understanding OCR information in text-rich images such as menus, tickets, bills, etc. For example, given a photo of a ticket, the sub-instruction asking about the price printed on the ticket falls into this category.

Figure 5 shows the most frequently used verbs and co-occurring nouns in MIA-Bench. To guarantee the diversity of prompts, when writing the instructions, we construct instructions of various levels of complexity: *basic*, *intermediate*, *advanced*, *creative*, and *complex*. The *basic* category is the simplest; the instructions normally only contain one or two sub-instructions, such as "What is the color of the cat?", or "Describe the sofa in two words.". The *intermediate* category consists of instructions that contain three or more sub-instructions, but are in general easy for MLLMs to follow. The *advanced* category contains instructions that are challenging and contain three or more sub-instructions. The *creative* category contains instructions that instruct MLLMs to generate creative pieces of text, such as poems. The *complex* category is a combination of the previous two categories; the instructions in this category are the most complicated as they usually contain multiple challenging sub-instructions. While we found these categories useful to elicit a diverse instruction set, we also found that practical examples were often difficult to categorize objectively. As a result, we only used these categories for data collection, but are not reporting per-category results.

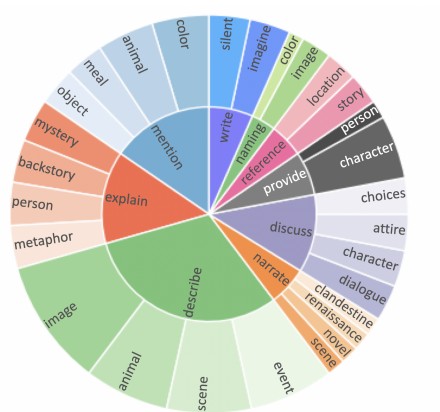

Figure 5: The most frequently used verbs and co-occurring nouns in MIA-Bench.

## 2.2 RESPONSE EVALUATION METHOD

To automatically evaluate MLLMs' performance with the proposed MIA-Bench at scale, we adopt GPT-4o (OpenAI, 2024) as a judge model to score MLLMs' responses on each instruction and return a total score based on different criteria mentioned above. Specifically, we design the following text prompt template:

> Here is the instruction for a multimodal LLM: {instruction} You need to grade if the response from the model follows each component of the instruction. The first component is: {sub-instruction 1}, and the second component is: {sub-instruction 2}, … , and the Nth component is: {sub-instruction N}. The response is: {response} You need to score the response and be strict. The total score ranges from 0 to 10, depending on if the response follows the instruction. The first, second, …, and Nth component is each worth {weight 1}, {weight 2}, …, and {weight N} scores. List scores of each component, and the total score in one sentence in this format: score of component 1: x/2, score of component 2: y/8, total score: z/10. Then explain

Each response is graded through a systematic process that begins by thoroughly evaluating how well it adheres to each specific sub-instruction provided within the overall task. This initial assessment focuses on the degree of compliance and alignment with the requirements outlined in the sub-instructions. After this detailed evaluation of individual components, a total score is computed by aggregating the performance across all sub-instructions. Each sub-instruction within the broader instruction is assigned a maximum possible score, which can range from 1 to 10. The specific score assigned to a sub-instruction reflects its relative significance and difficulty within the context of the overall task. To ensure fairness and consistency, the total weight of all sub-instructions for any given instruction is set to sum up to 10. This carefully calibrated scoring system has been meticulously designed to capture and reflect the varying levels of complexity and importance associated with each aspect of the task. By doing so, it ensures that more critical or challenging elements are appropriately weighted and that the evaluation process provides a comprehensive and accurate representation of the response's quality.

Figure 6 shows an example of how responses from different MLLMs are evaluated and scored. For example, length limits are often binary in nature—either met or not met—hence the single point allocation. In contrast, a description task may require the model to handle multiple layers of complexity, including accuracy, detail, and relevance, which justified a higher score. For the example in Figure 6, there are 4 sub-instructions (denoted from S1 to S4); the first is worth 4 points and the rest is worth 2 points each. The response from GPT-4o partially follows the first sub-instruction which requires the response to be from the perspective of the dog, as the dog should not be able to see the car behind the man without turning around. The dog should be able to see the guitar, thus GPT-4o

**Instruction:**

Imagine you are the dog in the image, list 2 objects you can see without moving your head. Respond in bullet points, within 20 words.

**Four Sub-instructions (10 points):**

S1: Answer from the dog's perspective, without moving its head, 4 points
S2: List 2 objects, 2 points
S3: Respond in bullet points, 2 points
S4: Limit response in 20 words, 2 points

**GPT-4o (8 points):**

• Guitar with a brown and white finish
• Car with a visible license plate
S1❌the dog can't see the car (2/4); S2✅ (2/2); S3✅ (2/2); S4✅ (2/2)

**InternVL-Chat-V1.5 (8 points):**

• The person holding me is playing a guitar.
• There's a black case in front of the person.
S1✅ (4/4); S2❌response contains 3 objects instead of 2 (0/2); S3✅ (2/2); S4✅ (2/2)

**XComposer2 (8 points):**

2 objects: the guitar and person playing it.
S1✅ (4/4); S2✅ (2/2); S3❌not in bullet points (0/2); S4✅ (2/2)

**LLaVA-1.6-34b (10 points):**

• Microphone stand
• Guitar
S1✅ (4/4); S2✅ (2/2); S3✅ (2/2); S4✅ (2/2)

Figure 6: An example featuring responses from four MLLMs along with their evaluation scores, presented in a point-by-point format.

gets 2 points out of 4 for the first sub-instruction. It successfully follows the other 3 sub-instructions, achieving full score for them. Thus, the final score GPT-4o reaches is 8 out of 10. We always assign larger weight (6 if there are two sub-instructions, 4 if there are three or more sub-instructions) to the sub-instruction in the *description* category unless this category is absent in some cases, as usually a major part of the response is addressing this sub-instruction. For each MLLM, we compute the average score it gets on all 400 responses, and represent the ratio of the average score divided by 10 using percentage. We also compute the average score for each instruction category.

# 3 EXPERIMENTS

In this section, we first present results of different MLLMs on MIA-Bench in Section 3.1, with additional supervised fine-tuning exploration in Section 3.2.

## 3.1 BENCHMARK RESULTS

In total, we have evaluated 29 popular MLLMs on MIA-Bench. Results are reported in Table 1. Observations are summarized as follows.

• Overall, the best performance was achieved by GPT-4o (OpenAI, 2024), with a score 88.58, showcasing its superiority across different categories of instruction adherence.

• The ability to describe content accurately was best exhibited by Reka (Team et al., 2024). Other models like Claude-3-Opus (Anthropic, 2024), GPT-4v (Achiam et al., 2023) and GPT-4o also achieved scores higher than 90. This suggests that these models are good at generating coherent and contextually appropriate text.

• In the genre category, the highest proficiency was shown by GPT-4v and GPT-4o with scores above 94, suggesting an exceptional grasp of language nuances. Among open-source models, Phi-3-Vision (Abdin et al., 2024) and LLaVA-NeXT-34b (Liu et al., 2024a) show strong performance with scores of 86.52 and 86.58, respectively. The lowest score on this metric was by Kosmos-2 (Peng et al., 2023a), with a mere 11.55, pointing to difficulties in understanding or generating linguistically complex sentences.

• GPT-4o excelled in grammar with a score of 85.70, which indicates superior ability in syntax correctness and sentence structuring that matches specific requirements in the instruction. Among the open-source models, MiniCPM-Llama3-V-2.5 (Hu et al., 2023) is notable with a score of

| Model | Meta-Avg | Description | Len-Limit | Genre | Grammar | Mention | Math | Perspective | OCR |
|---|---|---|---|---|---|---|---|---|---|
| Open Source 1b−8b | | | | | | | | | |
| Fuyu-8b (Bavishi et al., 2023) | 24.52 | 52.06 | 24.52 | 17.06 | 17.18 | 36.43 | 22.62 | 66.67 | 33.09 |
| Kosmos-2 (Peng et al., 2023a) | 26.06 | 50.95 | 38.52 | 11.55 | 19.78 | 28.70 | 17.26 | 50.83 | 41.88 |
| Sphinx (Lin et al., 2023b) | 50.99 | 75.33 | 53.51 | 60.45 | 48.28 | 57.75 | 47.41 | 70.00 | 61.04 |
| Idefics-2-8b (Laurençon et al., 2024) | 51.42 | 59.37 | 62.73 | 48.07 | 64.09 | 46.20 | 46.51 | 48.33 | 61.97 |
| mPLUG-Owl2 (Ye et al., 2023b) | 57.86 | 75.01 | 65.25 | 63.39 | 60.26 | 57.70 | 57.22 | 65.00 | 62.08 |
| CogVLM-Chat (Wang et al., 2023a) | 58.95 | 60.42 | 57.86 | 67.94 | 60.55 | 62.92 | 36.67 | 60.83 | 61.87 |
| ShareGPT4V (Chen et al., 2023b) | 59.41 | 81.08 | 63.49 | 63.88 | 58.46 | 62.49 | 52.98 | 82.50 | 72.29 |
| DeepSeek-VL-7b-chat (Lu et al., 2024) | 60.96 | 86.31 | 63.26 | 72.11 | 54.79 | 63.75 | 67.39 | 74.17 | 77.85 |
| LLaVA-1.5-7b (Liu et al., 2023c) | 62.18 | 78.00 | 68.60 | 63.95 | 64.18 | 65.89 | 47.31 | 86.67 | 60.75 |
| LLaVA-NeXT-7b-vicuna (Liu et al., 2024a) | 62.27 | 79.21 | 68.01 | 65.63 | 60.95 | 63.33 | 46.67 | 90.00 | 65.54 |
| Qwen-VL-Chat (Bai et al., 2023) | 63.09 | 80.51 | 74.22 | 66.95 | 63.11 | 63.01 | 45.00 | 75.83 | 66.01 |
| XComposer2-7b (Dong et al., 2024a) | 67.71 | 83.47 | 76.16 | 73.66 | 67.69 | 67.01 | 48.61 | 77.50 | 68.06 |
| CogVLM2 (Wang et al., 2023a) | 73.43 | 87.60 | 74.52 | 83.47 | 71.97 | 77.01 | 71.53 | 90.83 | 87.16 |
| Phi-3-vision (Abdin et al., 2024) | 76.02 | 84.90 | 84.46 | 86.52 | 67.93 | 74.70 | 78.16 | 74.17 | 83.96 |
| MiniCPM-Llama3-v2.5 (Hu et al., 2023) | 76.27 | 84.12 | 79.44 | 80.33 | 81.25 | 76.99 | 64.08 | 81.67 | 76.59 |
| Open Source 8b−13b | | | | | | | | | |
| InstructBLIP-13b (Dai et al., 2023) | 38.16 | 50.54 | 39.57 | 29.34 | 38.43 | 42.28 | 12.50 | 50.00 | 30.42 |
| LLaVA-1.5-13b (Liu et al., 2023c) | 63.55 | 80.98 | 70.15 | 64.54 | 59.30 | 67.42 | 45.11 | 69.17 | 76.28 |
| LLaVA-NeXT-13b-vicuna (Liu et al., 2024a) | 69.16 | 86.75 | 69.88 | 82.07 | 64.77 | 74.99 | 48.56 | 77.50 | 75.83 |
| Open Source 13b−110b | | | | | | | | | |
| Yi-VL-34b (AI et al., 2024) | 53.90 | 74.89 | 52.05 | 59.09 | 55.91 | 57.25 | 54.17 | 41.85 | 70.09 |
| InternVL-Chat-v1.5 (Chen et al., 2024b) | 75.42 | 89.13 | 78.21 | 79.92 | 78.16 | 77.54 | 76.11 | 87.50 | 80.92 |
| LLaVA-NeXT-34b (Liu et al., 2024a) | 75.61 | 88.02 | 83.50 | 86.58 | 71.57 | 75.83 | 68.06 | 87.50 | 80.26 |
| LLaVA-NeXT-110b (Liu et al., 2024a) | 79.84 | 86.99 | 84.86 | 82.49 | 79.04 | 80.10 | 71.94 | 80.83 | 75.45 |
| Proprietary | | | | | | | | | |
| Gemini-1.0-Pro (Team, 2023) | 70.63 | 82.77 | 72.83 | 78.76 | 76.91 | 71.67 | 81.45 | 89.29 | 84.11 |
| Reka-Core (Team et al., 2024) | 76.95 | **91.05** | 79.91 | 85.16 | 78.98 | 82.08 | 82.53 | 77.50 | 81.08 |
| Claude-3-Haiku (Anthropic, 2024) | 78.25 | 86.86 | 77.53 | 90.27 | 73.41 | 82.62 | 82.22 | 57.50 | 86.49 |
| Claude-3-Sonnet (Anthropic, 2024) | 79.44 | 88.06 | 82.71 | 90.54 | 79.60 | 82.05 | 82.22 | 76.67 | 84.43 |
| Claude-3-Opus (Anthropic, 2024) | 84.50 | 90.50 | 86.03 | 91.19 | 83.82 | 85.49 | 85.92 | 65.00 | **86.84** |
| GPT-4v (Achiam et al., 2023) | 86.11 | 90.03 | 87.61 | **94.59** | 80.12 | 89.37 | 85.63 | 59.17 | 85.26 |
| GPT-4o (OpenAI, 2024) | **88.58** | 90.82 | **92.73** | 94.29 | **85.70** | 90.66 | **87.07** | **92.50** | 86.54 |

Table 1: Evaluation results of a wide array of MLLMs on MIA-Bench.

81.25. Contrastingly, Fuyu-8b (Bavishi et al., 2023) scored the lowest with 17.18, reflecting major challenges in grammar adherence.

- GPT-4o also showed the best performance with a score of 92.73 in respecting prescribed length limits, which is crucial for tasks requiring concise and precise answers. Among open-source models, LLaVA-NeXT-110b (Liu et al., 2024a) stands out with a score of 84.86.

- Results from LLaVA series also suggest a strong correlation between LLM size and MIA-Bench performance across metrics.

**Correlation with other benchmarks.** In Table 2, we compare the ranking of 5 state-of-the-art MLLMs on MIA-Bench as well as their meta ranking on MME (Fu et al., 2023a), MMMU (Yue et al., 2023) , MMBench (Liu et al., 2024b), MMVet (Yu et al., 2023b), HallusionBench (Guan et al., 2023), and MathVista (Lu et al., 2023) (meta ranking is computed by averaging rankings across these benchmarks). Our findings reveal a discrepancy between the two sets of rankings. Notably, InternVL-Chat-V1.5 (Chen et al., 2024b), which holds the highest meta-ranking among the five MLLMs on the other benchmarks, ranks the lowest on MIA-Bench. Conversely, Claude-3-Opus, which has the lowest meta-ranking, secures the second position on the MIA-Bench. This indicates that excelling in tasks evaluated by existing benchmarks does not necessarily translate to superior instruction adherence capability assessed by MIA-Bench.

| Model | MME | MMMU | MMB | MMVet | HallB | Math Vista | Meta Ranking | MIA | MIA Ranking |
|---|---|---|---|---|---|---|---|---|---|
| GPT-4o (OpenAI, 2024) | 2328.7 | 69.1 | 83.3/82.1 | 66.5 | 67.5 | 63.8 | 1 | 88.58 | 1 |
| GPT-4v (Achiam et al., 2023) | 1926.6 | 56.8 | 77/74.4 | 67.6 | 46.5 | 49.9 | 3 | 86.11 | 2 |
| Gemini-1.0-Pro (Team, 2023) | 1933.4 | 47.9 | 73.6/74.3 | 64.3 | 45.2 | 45.2 | 5* | 70.63 | 6 |
| Claude-3-Opus (Anthropic, 2024) | 1586.8 | 59.4 | 63.3/59.2 | 58.1 | 37.8 | 50.5 | 5* | 84.50 | 3 |
| InternVL-Chat-V1-5 (Chen et al., 2024b) | 2187.8 | 45.2 | 82.2/82 | 62.8 | 49.3 | 53.5 | 2 | 75.42 | 5 |
| LLaVA-NeXT-34b (Li et al., 2024a) | 2028 | 51.1 | 81.1/79 | 48.9 | 47.6 | 47.7 | 4 | 75.61 | 4 |

Table 2: Meta ranking of five state-of-the-art MLLMs on existing multimodal benchmarks compared with their ranking on MIA-Bench.

**Correlation with LLM backbone performance.** To determine if the performance on MIA-Bench is attributable solely to the underlying LLMs, we also evaluate several MLLMs on IFEval (Qin et al., 2024), a benchmark that assesses the instruction adherence capability of LLMs, and compare their ranking with that on MIA-Bench. This comparison is shown in Figure 7, which shows that the instruction adherence capabilities of MLLMs do not consistently align with their LLMs' adherence capability.

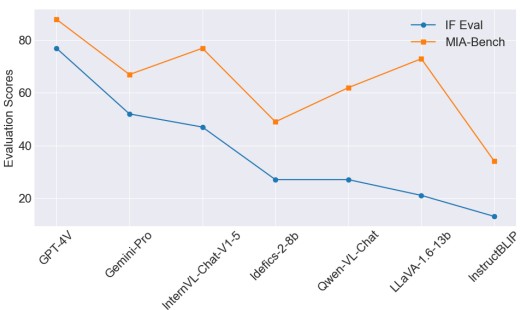

Figure 7: Study of correlation between scores on MIA-Bench and IFEval.

**Other external models as the judge.** Since the evaluation uses GPT-4o as the judge, it is natural to conjecture that GPT-4o may favorably score its own responses. To alleviate this concern, we use Claude-3, a strong performer in Table 1, to evaluate responses from GPT-4o and itself, and compare their scores with each other. The prompt used to grade responses is the same as the one used in GPT-4o grading. We find that even using Claude-3 Opus to score its own and GPT-4o's responses, GPT-4o still achieves a superior score. When scored by Claude-3-Opus, GPT-4o achieves 89.84 score in contrast to Claude-3-Opus' 85.89. Based on this observation, we use GPT-4o for evaluation by default, and observe that results from multiple runs may have around 1% variation.

We further extend our study by evaluating the models using various versions of ChatGPT as judge models, including gpt-4o-mini-2024-07-18, gpt-4o-2024-05-13, gpt-4o-2024-11-20, and chatgpt-4o-latest, and show the scores and ranking in Table 3 with detailed results in Table 6, 7, 8 and 9. We find that the ranking is consistent even though the judge models are different.

| Model | chatgpt-4o-latest | | gpt-4o-2024-11-20 | | gpt-4o-2024-05-13 | | gpt-4o-mini-2024-07-18 | |
|---|---|---|---|---|---|---|---|---|
| | Score | Ranking | Score | Ranking | Score | Ranking | Score | Ranking |
| GPT-4o | 89.69 | 1 | 89.94 | 1 | 90.97 | 1 | 81.36 | 1 |
| Claude-3-Opus | 86.16 | 2 | 84.89 | 2 | 85.61 | 2 | 78.95 | 2 |
| Reka | 83.09 | 3 | 82.68 | 3 | 83.99 | 3 | 77.70 | 3 |
| MiniCPM-Llama3-V2.5 | 78.10 | 4 | 78.75 | 4 | 79.80 | 4 | 73.72 | 4 |
| Gemini | 75.77 | 5 | 76.32 | 5 | 77.36 | 5 | 67.45 | 5 |
| LLaVA-1.5-13b | 66.78 | 6 | 66.05 | 6 | 68.39 | 7 | 61.54 | 6 |
| ShareGPT4v | 66.61 | 7 | 65.72 | 7 | 68.90 | 6 | 60.30 | 7 |
| Idefics-2-8b | 53.51 | 8 | 53.61 | 8 | 54.18 | 8 | 44.28 | 8 |

Table 3: Comparison of scores and rankings across different judge models. The ranking is stable.

## 3.2 SUPERVISED FINE-TUNING (SFT)

The performance of small-scale models such as LLaVA-NeXT-13b is less ideal on MIA-Bench. In this section, we study the use of supervised fine-tuning to enhance model performance.

**Additional SFT data construction.** First, we randomly sample 1000 images from COCO 2017 training set, and use GPT-4v to generate five instructions for each image, using the prompt below.

| Model | Total Score | Description | Length Limit | Genres | Grammar | Mention | Math | Perspective | OCR |
|---|---|---|---|---|---|---|---|---|---|
| LLaVA-NeXT-13b (Liu et al., 2024a) | 69.16 | 86.75 | 69.88 | 82.07 | 64.77 | 74.99 | 48.56 | 77.50 | 75.83 |
| + DIT | 78.85 | 86.90 | 86.80 | 88.02 | 71.34 | 81.01 | 60.87 | 84.17 | 72.65 |
| + DIT + LLaVA-Instruct150k | 78.90 | 88.59 | 74.67 | 79.95 | 74.17 | 66.39 | 53.70 | 100.00 | 80.83 |

Table 4: Detailed results of LLaVA-NeXT-13b (Liu et al., 2024a) on MIA-Bench before and after supervised fine-tuning on additional constructed diverse instruction-tuning (DIT) data, and the mixture of DIT and LLaVA-Instruct150k. We re-ran the baseline.

| Model | MMBench | TextVQA | VQA2 | LLaVA-itw | POPE | VizWiz | MIA-Bench |
|---|---|---|---|---|---|---|---|
| LLaVA-NeXT-13b (Liu et al., 2024a) | 70.6 | 64.26 | 82.80 | 85.8 | 87.7 | 60.41 | 69.16 |
| + DIT | 68.6 | 63.20 | 82.58 | 83.4 | 86.9 | 59.72 | 78.85 |
| + DIT + LLaVA-Instruct150k | 67.27 | 54.24 | 77.92 | 75.8 | 87.8 | 58.87 | 78.90 |

Table 5: Results of LLaVA-NeXT-13b (Liu et al., 2024a) on MIA-Bench and other major multimodal benchmarks supervised fine-tuning on additional constructed diverse instruction-tuning (DIT) data, and the mixture of DIT and LLaVA-Instruct150k. We re-ran the baseline.

> Sometimes multimodal LLM doesn't follow user's instruction. For example, when I ask a model to use three sentences to answer my question, it might answer in four sentences. I want to evaluate models' ability to strictly follow instructions. Help me compose instructions that are of five levels of difficulty to follow, for this image.
> The five levels are 'Basic Instruction, Intermediate Instruction, Advanced Instruction, Creative Instruction, Complex Instruction'. Diversify when composing instructions. After each of your instructions, add ***.

We then manually process the generated instructions. The cleaned data for SFT consists of 5000 image-prompt pairs.

Then, we use GPT-4v to generate responses to the constructed prompts. To evaluate the quality of these responses, we sampled 100 responses and manually checked if they adhere to the instructions. We find that 90% of the responses successfully followed all instructions in the prompt, serving as a proper ground-truth response for model training. Examples of this additional training data is provided in the Appendix.

**Results.** Using LLaVA-NeXT-13b as the backbone, we train the model for 1 epoch on the constructed diverse instruction-tuning (DIT) data. We also performed SFT using the combination of LLaVA Visual Instruct 150K dataset and our diverse instruction-tuning dataset, to examine which data mixture leads to better performance. Results on MIA-Bench and other benchmarks are summarized in Table 5, with detailed results on MIA-Bench reported in Table 4. The performance of the model after finetuning on the DIT data alone on MIA-Bench has been significantly improved by around 10 points, at the cost of minor regressions across other benchmarks. This supports our hypothesis that using the small amount of DIT data, we can enhance the model with better instruction following capability. On the other hand, further adding LLaVA Visual Instruct 150K data did not significantly improve model performance on the tested benchmarks. Examples are shown in Figure 8 to compare responses from LLaVA-NeXT-13b before and after SFT.

# 4 RELATED WORK

**Multimodal LLMs and Benchmarks.** Multimodal Large Language Models (MLLMs) have recently emerged as a significant research focus. LLaVA (Liu et al., 2023c) and MiniGPT-4 (Zhu et al., 2023) pioneered visual instruction tuning, and the past year has witnessed a boom of open-source MLLMs based on this concept. Prominent examples include InstructBLIP (Dai et al., 2023), mPLUG-Owl(-2/Doc) (Ye et al., 2023b;c;a), Qwen-VL (Bai et al., 2023), CogVLM (Wang et al., 2023a), SPHINX(-X) (Lin et al., 2023b; Gao et al., 2024), InternLM-XComposer2-VL (Dong et al., 2024b), InternVL(-1.5) (Chen et al., 2023c; 2024b), VILA (Lin et al., 2023a), MM1 (McKinzie et al., 2024), Mini-Gemini (Li et al., 2024b), Idefics2 (Laurençon et al., 2024), Phi-3-vision (Abdin et al., 2024), to name a few. There is also a rich body of literature on enabling MLLMs for referring and grounding (Peng et al., 2023b; Chen et al., 2023a; You et al., 2024a; Wang et al., 2023b; Lai et al., 2023; Zhang et al., 2023; 2024; You et al., 2024b), image generation and editing (Koh et al., 2023; Sun et al., 2023a; Fu et al., 2023b), *etc*.

Various benchmarks have been proposed to evaluate the performance of MLLMs across different dimensions. Benchmarks like VQAv2 (Goyal et al., 2017), TextVQA (Singh et al., 2019a), Sci-

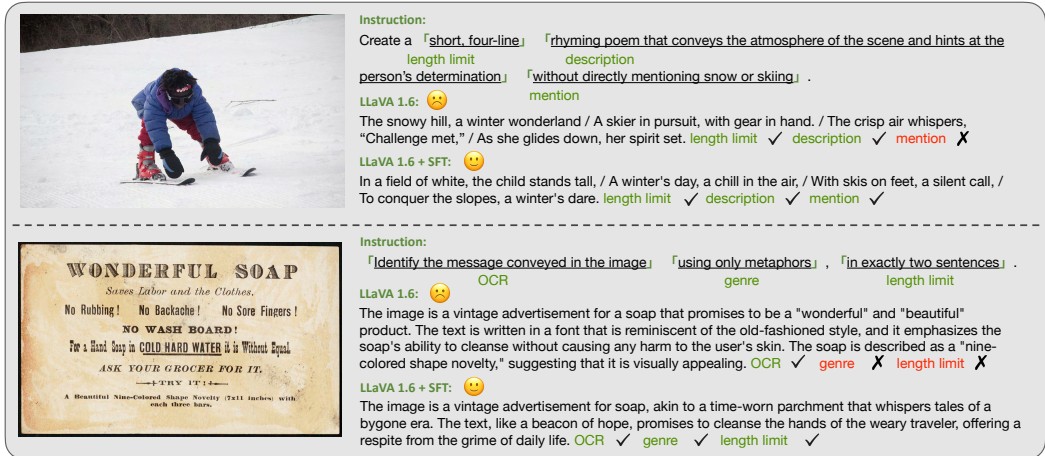

Figure 8: Examples of LLaVA-NeXT-13b responses before and after supervised fine-tuning on additional diverse instruction-tuning data.

enceQA (Lu et al., 2022), MME (Fu et al., 2023a), MMbench (Liu et al., 2024b), SEED-Bench (Li et al., 2023a), MathVista (Lu et al., 2023), and MMMU (Yue et al., 2023) aim to assess comprehensive multimodal understanding abilities. Additionally, there are benchmarks that specifically study model hallucination, including POPE (Li et al., 2023b), MHalDetect (Gunjal et al., 2024), GAVIE (Liu et al., 2023a), HallusionBench (Guan et al., 2023), and MAD-Bench (Qian et al., 2024). Many of these benchmarks have gained popularity within the community due to their use of multiple-choice evaluations. However, they do not accurately reflect the common use cases for MLLMs, where user interactions are typically open-ended. To address this, benchmarks like LLaVA-Bench (Liu et al., 2023c), MM-Vet (Yu et al., 2023b), and Vibe-Eval (Padlewski et al., 2024) have been proposed. Our MIA-Bench also falls into this category; however, we focus on studying the exact instruction adherence of MLLMs, a metric that previous benchmarks have only loosely measured.

**Instruction Following Benchmarks for LLMs.** Several benchmarks have been proposed to measure the instruction adherence ability of LLMs. Instruction-Following Eval (IFEval) (Zhou et al., 2023) is a benchmark for assessing LLMs' adherence ability to the given instructions. Its approach emphasizes verifiable instructions, which enhance objectivity and reproducibility in evaluations. IFEval creates 541 prompts spanning 25 instruction types, revealing a significant performance gap in instruction adherence ability between GPT-4 (Achiam et al., 2023) and PaLM-2 (Anil et al., 2023). This demonstrates the benchmark's ability to effectively differentiate between models in adherence ability. On the other hand, InfoBench (Qin et al., 2024) introduces a new metric called Decomposed Requirements Following Ratio (DRFR) for assessing the instruction-adherence capabilities of LLMs. DRFR dissects complex instructions into simpler sub-instructions, allowing for a granular evaluation of compliance with various task aspects. InfoBench contains 500 diverse instructions consisting of 2,250 decomposed questions in multiple constraint categories. The evaluation of advanced LLMs using this framework highlights their strengths and areas for improvement, especially in complex instruction adherence scenarios. Compared with these previous work, we are the first known effort that specifically focuses on benchmarking the instruction adherence ability of multimodal LLMs.

## 5 CONCLUSION

This paper introduces MIA-Bench, a benchmark designed to evaluate the ability of MLLMs to strictly adhere to complex instructions within prompts. Through the analysis of 400 image-prompt pairs from diverse sources, our findings highlight variability in model performance and much room for improvement, underscoring a critical need for enhanced training methods to improve instruction compliance. We further explored supervised fine-tuning (SFT) using LLaVA-NeXT as the backbone, which yielded promising results. Going forward, future research can expand on both SFT and alignment methods such as RLHF (Ouyang et al., 2022; Sun et al., 2023b) and DPO (Rafailov et al., 2024; Yu et al., 2023a), enhancing MLLMs to achieve higher accuracy and reliability in practical applications across diverse instructional contexts.

## LIMITATION

In designing the instructions for our benchmark, we incorporated a wide range of categories to enhance the diversity of sub-instructions. Nonetheless, the real world presents an infinite variety of instructions, many of which may pose significant challenges for MLLMs.

## REPRODUCIBILITY STATEMENT

For reproducibility purpose, we release our evaluation code and benchmark at: `https://github.com/apple/ml-mia-bench`.

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

## A Appendix

### A.1 Examples of diverse instruction-tuning data used for SFT

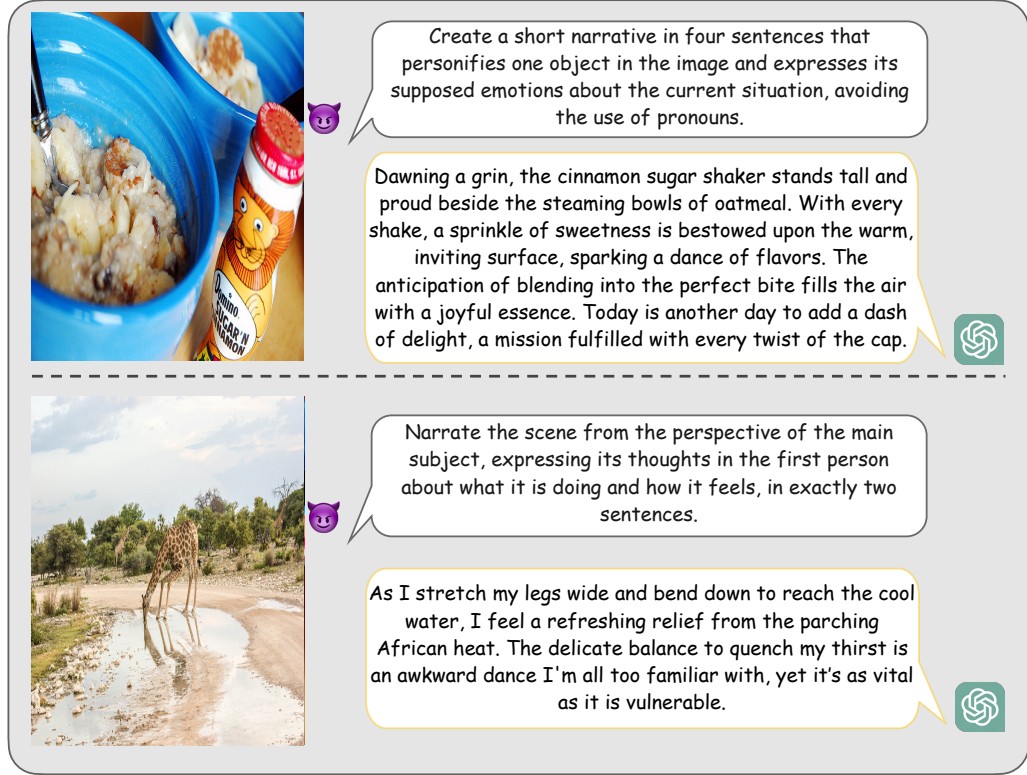

Figure 9: Examples of diverse instruction-tuning data used for SFT.

### A.2 Examples of how MLLMs respond to instructions in MIA-Bench

**Instruction:**
Imagine a short story of two sentences about why the animal is in its current location, but you must include a weather condition, a time of day, and an emotion. components: ["Imagine a short story about why the animal is in its current location", "of two sentences", "you must include a weather condition, a time of day, and an emotions"], component weight: [4, 3, 3]

**XComposer2-7b:**
On a sunny afternoon, the polar bear, feeling relaxed, lounges on a rock in its enclosure, enjoying the warmth of the sun.
Score of component 1: 4/4, score of component 2: 0/3, score of component 3: 3/3, total score: 7/10.
Reasons: Component 1: The story explains why the polar bear is in its current location (lounging on a rock in its enclosure enjoying the warmth of the sun). Full score awarded. Component 2: The response is not two sentences long. Component 3: The story includes a weather condition (sunny), a time of day (afternoon), and an emotion (relaxed). Full score awarded.

**LLaVA-NeXT 110b:**
The polar bear is lounging on the rocks by the pool, enjoying the warmth of the sun on its fur. It's a beautiful day at the zoo, and the bear seems content in its enclosure.
Score of component 1: 3/4, score of component 2: 3/3, score of component 3: 2/3, total score: 8/10.
Reasons: Component 1 (short story about why the animal is in its current location): The response provides context about why the polar bear is in its current location (lounging by the pool on a beautiful day at the zoo). However, it could be more detailed in explaining why it's there specifically. Component 2 (two sentences): The response is effectively two sentences. Component 3 (weather condition, time of day, and emotion): The response includes a weather condition (warmth of the sun, beautiful day), a time of day (implicitly daytime implied by the sun), and an emotion (content). However, the time of day could be made more explicit.

**Qwen-VL-Chat:**
The polar bear is resting on the rock in the sun because it's a warm summer day. It feels relaxed and comfortable in its natural habitat.
Score of component 1: 4/4, score of component 2: 3/3, score of component 3: 2/3, total score: 9/10.
Reasons: Component 1: The story does explain why the polar bear is in its current location. Component 2: The response is exactly two sentences in length, fully meeting this requirement. Component 3: The response includes a weather condition ("warm summer day") and an emotion ("relaxed and comfortable"). However, it does not clearly specify a time of day, losing 1 point.

**GPT-4v:**
Under the harsh midday sun, the polar bear lounged by the water's edge, seeking relief from the sweltering heat. Despite the warmth, a glimmer of contentment shone in its eyes as it enjoyed a moment of quiet repose.
Score of component 1: 4/4, score of component 2: 3/3, score of component 3: 3/3, total score: 10/10.
Reasons: Component 1: The response explains why the polar bear is at the water's edge, seeking relief from the heat. Full marks. Component 2: The response has exactly two sentences. Full marks. Component 3: The response includes a weather condition (harsh midday sun), a time of day (midday), and an emotion (contentment). Full marks.

Figure 10: An example of how MLLMs respond to instructions in MIA-Bench.

**Instruction:**
Explain the message conveyed by the text on the sign in the image, using exactly two sentences.
components: ["Explain the message conveyed by the text on the sign in the image", "using exactly two sentences"], component weight: [6, 4]

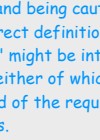

**InternVL-Chat-V1-5:**
The text on the sign in the image conveys a message of stopping and being cautious, as it reads "STOP" and "VOLDEMICKY." The word "VOLDEMICKY" is a play on the word "volde-mekky," which is a Dutch term for a person who is easily influenced or manipulated. The message suggests that one should be careful not to be easily swayed or manipulated by others.
Score of component 1: 3/6, score of component 2: 0/4, total score: 3/10.
Reason: Component 1: The response explains the message of stopping and being cautious, but it misinterprets "VOLDEMICKY" as "volde-mekky" and provides an incorrect definition. Additionally, there seems to be a misunderstanding as "VOLDEMICKY" might be intended as a playful reference to a combination of Voldemort and Mickey Mouse, neither of which is explained.\n- Component 2: The response uses three sentences instead of the required two, hence it does not follow the instruction of using exactly two sentences.

**Claude Opus:**
The text on the illuminated stop sign conveys an emphatic command to halt or cease movement. The unusual octagonal shape and red color of the sign reinforce the urgent, attention-grabbing nature of the message to stop immediately.
Score of component 1: 4/6, score of component 2: 4/4, total score: 8/10.
Reason: The response explains the message conveyed by the text on the stop sign effectively, noting both the command to stop and the reinforcement by the shape and color of the sign. However, it doesn't address the text "Voldemort" beneath the word "STOP," which is a crucial part of the message. The response uses exactly two sentences, fulfilling the second component perfectly.

**Gemini:**
The sign says "Stop Voldemort!" It is a reference to the Harry Potter series, in which Voldemort is the main antagonist.
Score of component 1: 6/6, score of component 2: 4/4, total score: 10/10.
Reason: The response successfully explains the message conveyed by the text on the sign, which is to "Stop Voldemort" and relates it to the Harry Potter series where Voldemort is the main antagonist. Additionally, the response adheres to the requirement of using exactly two sentences. Therefore, it fulfills both components of the instruction fully.

**GPT-4v:**
The sign is a modification of the traditional stop sign, introducing the words "Stop Violence." This is a call to action to end violence and can be interpreted as a public message advocating for peace and safety.
Score of component 1: 2/6, score of component 2: 4/4, total score: 6/10.
Reason: The response does follow the second component perfectly by using exactly two sentences. However, it does not correctly explain the message on the sign. The sign in the image has the text "Stop Voldemort," not "Stop Violence," indicating a message related to a fictional character rather than a general call to end violence. Hence, the response accurately addresses the number of sentences but fails significantly in conveying the correct message from the sign.

Figure 11: Another example of how MLLMs respond to instructions in MIA-Bench.

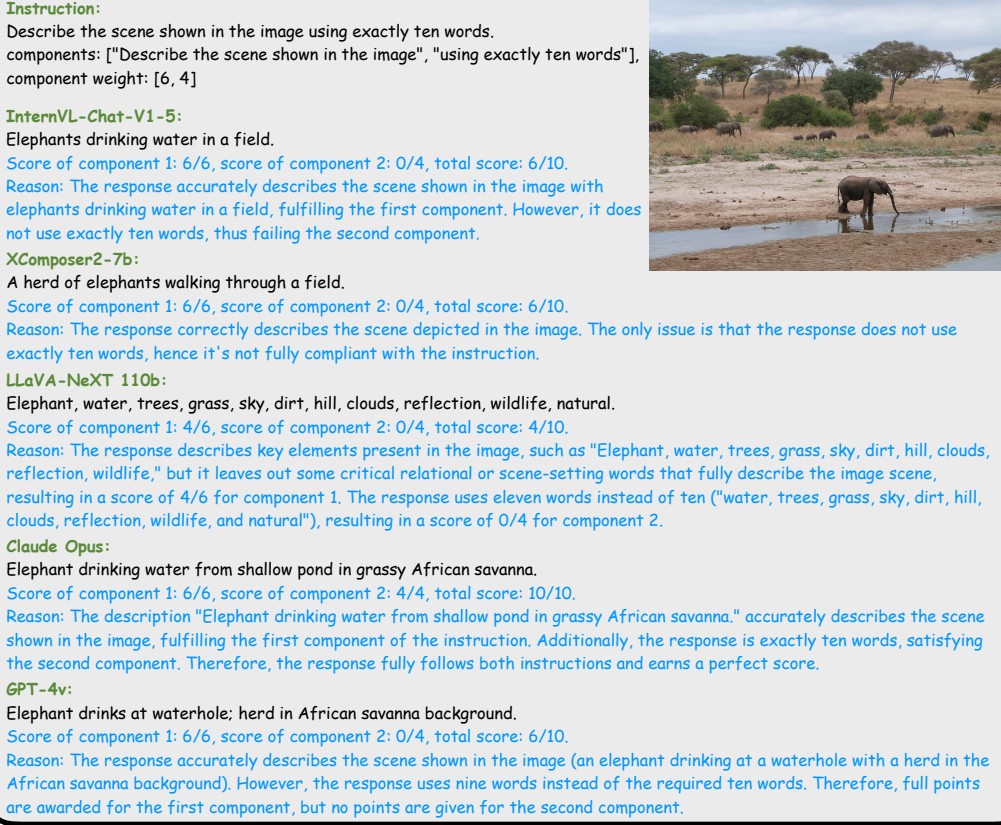

**Instruction:**
Describe the scene shown in the image using exactly ten words.
components: ["Describe the scene shown in the image", "using exactly ten words"],
component weight: [6, 4]

**InternVL-Chat-V1-5:**
Elephants drinking water in a field.
Score of component 1: 6/6, score of component 2: 0/4, total score: 6/10.
Reason: The response accurately describes the scene shown in the image with
elephants drinking water in a field, fulfilling the first component. However, it does
not use exactly ten words, thus failing the second component.

**XComposer2-7b:**
A herd of elephants walking through a field.
Score of component 1: 6/6, score of component 2: 0/4, total score: 6/10.
Reason: The response correctly describes the scene depicted in the image. The only issue is that the response does not use
exactly ten words, hence it's not fully compliant with the instruction.

**LLaVA-NeXT 110b:**
Elephant, water, trees, grass, sky, dirt, hill, clouds, reflection, wildlife, natural.
Score of component 1: 4/6, score of component 2: 0/4, total score: 4/10.
Reason: The response describes key elements present in the image, such as "Elephant, water, trees, grass, sky, dirt, hill, clouds,
reflection, wildlife," but it leaves out some critical relational or scene-setting words that fully describe the image scene,
resulting in a score of 4/6 for component 1. The response uses eleven words instead of ten ("water, trees, grass, sky, dirt, hill,
clouds, reflection, wildlife, and natural"), resulting in a score of 0/4 for component 2.

**Claude Opus:**
Elephant drinking water from shallow pond in grassy African savanna.
Score of component 1: 6/6, score of component 2: 4/4, total score: 10/10.
Reason: The description "Elephant drinking water from shallow pond in grassy African savanna." accurately describes the scene
shown in the image, fulfilling the first component of the instruction. Additionally, the response is exactly ten words, satisfying
the second component. Therefore, the response fully follows both instructions and earns a perfect score.

**GPT-4v:**
Elephant drinks at waterhole; herd in African savanna background.
Score of component 1: 6/6, score of component 2: 0/4, total score: 6/10.
Reason: The response accurately describes the scene shown in the image (an elephant drinking at a waterhole with a herd in the
African savanna background). However, the response uses nine words instead of the required ten words. Therefore, full points
are awarded for the first component, but no points are given for the second component.

Figure 12: The third example of how MLLMs respond to instructions in MIA-Bench.

## A.3 COMPARISON OF SCORES AND RANKINGS ACROSS DIFFERENT JUDGE MODELS

| Model | Total Score | Description | Length Limit | Genres | Grammar | Mention | Math | Perspective | OCR |
|---|---|---|---|---|---|---|---|---|---|
| GPT-4o | 0.813587 | 0.823910 | 0.847312 | 0.965594 | 0.733660 | 0.820370 | 0.775862 | 0.733333 | 0.765351 |
| Clause-3-Opus | 0.789474 | 0.805156 | 0.794086 | 0.916667 | 0.697115 | 0.771364 | 0.706897 | 0.629630 | 0.692308 |
| Reka | 0.776965 | 0.805536 | 0.807151 | 0.857904 | 0.732143 | 0.740253 | 0.687500 | 0.600000 | 0.689189 |
| MiniCPM-Llama3-V2.5 | 0.737190 | 0.715403 | 0.799068 | 0.849505 | 0.735294 | 0.711585 | 0.637931 | 0.716667 | 0.693694 |
| Gemini-1.0-Pro | 0.674504 | 0.676852 | 0.764286 | 0.768350 | 0.697695 | 0.615801 | 0.683333 | 0.611111 | 0.588235 |
| LLaVA-1.5-13b | 0.615427 | 0.663542 | 0.643424 | 0.643873 | 0.423077 | 0.619357 | 0.359195 | 0.666667 | 0.449561 |
| ShareGPT4v | 0.602989 | 0.674028 | 0.601496 | 0.625556 | 0.538462 | 0.560248 | 0.456897 | 0.555556 | 0.547619 |
| Idefics-2-8b | 0.442778 | 0.437893 | 0.494687 | 0.461218 | 0.520408 | 0.436042 | 0.276786 | 0.458333 | 0.412162 |

Table 6: Details of model scores evaluated by gpt-4o-mini-2024-07-18.

| Model | Total Score | Description | Length Limit | Genres | Grammar | Mention | Math | Perspective | OCR |
|---|---|---|---|---|---|---|---|---|---|
| GPT-4o | 0.909704 | 0.927875 | 0.912371 | 0.942057 | 0.862434 | 0.900441 | 0.857143 | 0.916667 | 0.882883 |
| Clause-3-Opus | 0.856077 | 0.890721 | 0.868490 | 0.917070 | 0.774590 | 0.819386 | 0.861111 | 0.725000 | 0.809524 |
| Reka | 0.839905 | 0.894752 | 0.785088 | 0.905643 | 0.713661 | 0.801667 | 0.925926 | 0.657407 | 0.828571 |
| MiniCPM-Llama3-V2.5 | 0.798023 | 0.828916 | 0.771795 | 0.823087 | 0.751944 | 0.763976 | 0.721264 | 0.816667 | 0.841880 |
| Gemini-1.0-Pro | 0.773569 | 0.817422 | 0.735470 | 0.788911 | 0.797814 | 0.683020 | 0.866071 | 0.870370 | 0.806373 |
| LLaVA-1.5-7b | 0.683947 | 0.758817 | 0.703750 | 0.674046 | 0.630208 | 0.617620 | 0.425287 | 0.800000 | 0.602564 |
| ShareGPT4v | 0.689046 | 0.800461 | 0.657738 | 0.608733 | 0.654762 | 0.601754 | 0.500000 | 0.800000 | 0.743056 |
| Idefics-2-8b | 0.541755 | 0.560243 | 0.619318 | 0.489276 | 0.646825 | 0.455342 | 0.405556 | 0.375000 | 0.627193 |

Table 7: Details of model scores evaluated by gpt-4o-2024-05-13.

| Model | Total Score | Description | Length Limit | Genres | Grammar | Mention | Math | Perspective | OCR |
|---|---|---|---|---|---|---|---|---|---|
| GPT-4o | 0.899410 | 0.909379 | 0.916204 | 0.969395 | 0.854885 | 0.861247 | 0.920290 | 0.907407 | 0.878378 |
| Clause-3-Opus | 0.848949 | 0.861543 | 0.871686 | 0.896552 | 0.797170 | 0.808777 | 0.846154 | 0.645833 | 0.865741 |
| Reka | 0.826844 | 0.881841 | 0.809259 | 0.873276 | 0.725309 | 0.770225 | 0.814103 | 0.750000 | 0.819444 |
| MiniCPM-Llama3-V2.5 | 0.787537 | 0.818813 | 0.790246 | 0.795796 | 0.768182 | 0.736359 | 0.676667 | 0.716667 | 0.828125 |
| Gemini-1.0-Pro | 0.763240 | 0.814379 | 0.750000 | 0.785159 | 0.757682 | 0.672255 | 0.758333 | 0.785714 | 0.776042 |
| LLaVA-1.5-7b | 0.660472 | 0.751873 | 0.661822 | 0.649851 | 0.498512 | 0.572719 | 0.516667 | 0.750000 | 0.571429 |
| ShareGPT4v | 0.657186 | 0.765309 | 0.632682 | 0.575578 | 0.583333 | 0.545104 | 0.464286 | 0.675000 | 0.717742 |
| Idefics-2-8b | 0.536134 | 0.589964 | 0.541887 | 0.455882 | 0.611582 | 0.449821 | 0.406667 | 0.527778 | 0.576190 |

Table 8: Details of model scores evaluated by gpt-4o-2024-11-20.

| Model | Total Score | Description | Length Limit | Genres | Grammar | Mention | Math | Perspective | OCR |
|---|---|---|---|---|---|---|---|---|---|
| GPT-4o | 0.896893 | 0.906288 | 0.917996 | 0.955952 | 0.830508 | 0.867949 | 0.846667 | 0.833333 | 0.896396 |
| Claude-3-Opus | 0.861628 | 0.895363 | 0.866039 | 0.927730 | 0.807018 | 0.820549 | 0.857639 | 0.666667 | 0.800926 |
| Reka | 0.830885 | 0.869867 | 0.821685 | 0.883403 | 0.795597 | 0.772894 | 0.813218 | 0.675000 | 0.848485 |
| MiniCPM-Llama3-V2.5 | 0.780966 | 0.831197 | 0.766026 | 0.796257 | 0.726190 | 0.722037 | 0.691358 | 0.656250 | 0.768018 |
| Gemini-1.0-Pro | 0.757733 | 0.793860 | 0.724138 | 0.745455 | 0.854167 | 0.670349 | 0.810606 | 0.822917 | 0.822581 |
| LLaVA-1.5-7b | 0.667826 | 0.743137 | 0.638889 | 0.675287 | 0.571212 | 0.594505 | 0.500000 | 0.758333 | 0.596774 |
| ShareGPT4v | 0.666092 | 0.773905 | 0.661290 | 0.573904 | 0.562500 | 0.570722 | 0.458333 | 0.638889 | 0.695238 |
| Idefics-2-8b | 0.535057 | 0.597963 | 0.531810 | 0.483768 | 0.593056 | 0.452361 | 0.326087 | 0.458333 | 0.569444 |

Table 9: Details of model scores evaluated by chatgpt-4o-latest.

