# OpenReview forum: "MIA-Bench: Towards Better Instruction Following Evaluation of Multimodal LLMs"
_ICLR.cc/2025/Conference — ICLR 2025 Poster_

### Official Review · Reviewer_UieD · 2024-11-03

**Soundness:** 2
**Presentation:** 3
**Contribution:** 2
**Rating:** 6
**Confidence:** 5

**Summary:**

The paper introduces MIA-Bench, a benchmark for evaluating the ability of multimodal large language models (MLLMs) to follow complex instructions. The benchmark consists of 400 image-prompt pairs that test the models' compliance with layered instructions to generate accurate responses. The evaluation of various state-of-the-art MLLMs shows significant performance variations, indicating areas for improvement in instruction adherence. The paper also discusses the creation of additional training data and supervised fine-tuning to enhance the models' instruction-following capabilities without affecting their performance on other tasks.

**Strengths:**

+ The paper presents a new benchmark that rigorously tests multimodal large language models' ability to follow complex instructions, addressing a previously underexplored area.

+ This paper offers a comprehensive set of 400 diverse image-prompt pairs that challenge models with layered instructions, enhancing the assessment of their linguistic and descriptive capabilities.

+ Experiment demonstrates performance improvements in instruction adherence through supervised fine-tuning.

**Weaknesses:**

- Complex instruction flowing ability might not be the main interests for MLLM users currently, especially when it comes to writting style constraints such as length and genre. Such abilities are mostly evaluated on LLMs.

- The performance of models on MIA-Bench does not necessarily correlate with their performance on other benchmarks, suggesting that excelling in this specific task may not translate to generalized improvements across different multimodal tasks.

**Questions:**

- In MIA-Bench, are instructions all related to the corresponding image content or randomly picked from a instruction bank?
- Instruction following abilities highly depends on LLMs used in the MLLM models. It would be better to group the performances in Table 1 and Table 2 according to the LLMs' size.

---

> ### Author Response · Authors · 2024-11-21
> **Response to Reviewer UieD**
>
> Thank you for your thoughtful review of our submission. We appreciate your feedback on both the strengths and areas for improvement in our paper. Below, we provide clarifications to your concerns.
>
> >Complex Instruction Following Relevance for MLLM Users
>
> We understand your concern regarding the importance of complex instruction-following capabilities, especially with respect to writing style constraints like length and genre. However, we argue that instruction adherence is increasingly important for MLLMs as they are used in applications requiring precise, multimodal interaction, such as visual-based assistants, educational tools, and creative applications where adherence to complex instructions (even stylistic) enhances user experience. For example, in educational scenarios, such as automated tutors, a teacher might instruct a model to "Summarize the image content as a short story suitable for a 7-year-old, incorporating a cheerful tone." Here, the combination of age-appropriate language, a specific tone, and multimodal comprehension tests the model’s ability to follow complex instructions critical for engaging and effective learning experiences. Multimodal cooking assistants could be tasked with, "Generate a recipe based on this image of ingredients, ensuring the instructions are concise, use metric measurements, and fit within a single screen view." Adherence to format, length, and context-specific guidelines enhances usability for users in real-time cooking environments. The MIA-Bench aims to address this need by focusing on layered and multifaceted instruction adherence tasks, which evaluate MLLMs beyond basic visual recognition, for more user-aligned multimodal interactions.
>
> >Performance Correlation with Other Benchmarks
>
> While MIA-Bench performance may not correlate directly with general-purpose benchmarks, this result aligns with our paper’s goal of highlighting instruction adherence as a distinct and specialized capability for MLLMs. This is our strength and motivation. Our findings show that excelling in MIA-Bench, which focuses on instruction adherence, serves as a unique measure of MLLMs’ capability that is not fully covered by most popular benchmarks.
>
> >Are instructions all related to the corresponding image content or randomly picked from a instruction bank?
>
> They are all related to the corresponding image, and MLLMs should not be able to provide a completely correct response without seeing the image. Each instruction is unique, written by human, and not picked from an instruction bank.
>
> >It would be better to group the performances in Table 1 and Table 2 according to the LLMs' size.
>
> Yes your suggestion makes a lot of sense. We have updated the paper to group open-source models into three categories: those with fewer than or equal to 8B parameters, those with 8B to 13B parameters, and those with more than 13B parameters.
>
> We sincerely appreciate your valuable feedback, which will contribute to enhancing the quality and impact of our work.

---

> > ### Comment · Reviewer_UieD · 2024-11-27
> >
> > Thanks for your feedback. Writting instructions accordingly and uniquely for each image makes this dataset helpful and valuable for the community. I have raised my rating for that.

---

### Official Review · Reviewer_PupP · 2024-11-04

**Soundness:** 3
**Presentation:** 3
**Contribution:** 3
**Rating:** 6
**Confidence:** 4

**Summary:**

To evaluate and guide the instruction-following capabilities of multimodal models, MIA-Bench introduces a set of benchmarks for multimodal instruction. MIA-Bench primarily measures the abilities of MLLMs on following layered and compositional multimodal instructions and provides a set of multimodal instructions to enhance model performance in these areas.

**Strengths:**

1. The paper proposes a benchmark for evaluating the abilities of MLLMs on following compositional instructions, covering a variety of categories and tasks, and providing guidance for MLLMs in following complex composite instructions in the future.

2. The paper proposes a set of instructions that can effectively enhance the ability of MLLMs to follow layered and compositional instructions.

**Weaknesses:**

1. Since the images in MIA-Bench are sourced from widely used datasets such as COCO 2017, SBU, TextVQA, and Flickr, I believe you should prioritize evaluating whether the current open-source models exhibit any data leakage issues on these datasets to demonstrate that MIA-Bench does not suffer from data contamination.

2.  I notice that the experiments in Table 3 were conducted on LLaVA-Next-13b. Is this done after fine-tuning on a preference-aligned model? If so, I think this continued SFT setup leading to a performance improvement in MIA-Bench, while other benchmarks exhibit varying degrees of decline, is quite intuitive. Could you please provide additional experiments on the impact of mixing generated instructions with original instructions on model performance? Furthermore, I notice that there are only 5000 generated instructions here; how would the model performance on MIA-Bench be affected if the number of instructions were increased?

**Questions:**

1. The 400 instructions in MIA Bench are manually annotated. It is important to ensure the reasonableness of the sub-instructions, such as whether the length limitation is appropriate. Additionally, has there been any manual verification of the scoring results from GPT-4o to confirm their accuracy and reasonableness?

2. The instructions in MIA-Bench are composed of multiple sub-instructions. How is the number of sub-instructions per instruction determined? Is there a difficulty grading for the instructions, such that a higher number of sub-instructions indicates a greater level of difficulty?

---

### Official Review · Reviewer_yNxN · 2024-11-08

**Soundness:** 3
**Presentation:** 3
**Contribution:** 2
**Rating:** 6
**Confidence:** 4

**Summary:**

This paper proposes MIA-Bench, which mainly focuses on the instruction-following capability of current Large Multimodal Models (LMMs) under complex and compositional instructions. Unlike previous close-ended benchmarks (e.g., multiple choice for MMBench) and open-ended benchmarks (e.g., LLaVA-Bench), MIA-Bench aims to evaluate precise adherence to complex instructions. As for evaluation, the authors leverage GPT-4o as the judge model. The authors have evaluated many representative LMMs on the proposed MIA-Bench.

**Strengths:**

1. Evaluating the capability of LMMs to adhere to complex instructions is worth studying.
2. Table 2 demonstrates that MIA-Bench is not highly correlated with existing benchmarks.
3. Figure 8 illustrates that MIA-Bench is vision-centric, which is not strongly correlated with the performance of the LLM backbone.

**Weaknesses:**

In the context of evaluating open-ended freeform responses like MIA-Bench, it is crucial to account for the inherent variability introduced by the judge model. For example, the performance of MM-Vet has been observed to fluctuate by up to 10 points when assessed with different versions of GPT.

This variability raises several important considerations:

1. **Standard Deviation Reporting:** When evaluating Large Multimodal Models (LMMs) using the same judge model (e.g., GPT-4o) across multiple trials, it is essential to report the standard deviation of the performance metrics.
2. **Detailed Generation Configuration:** The specific generation parameters of the judge model, such as top-p, top-k, temperature, and num_beams, should be explicitly documented.
3. **Impact of Generation Configuration:** It is necessary to investigate whether the above generation configuration of the judge model has a substantial impact on the performance metrics.
4. **Performance Across Different Judge Models:** Detailed performance metrics should be reported for various judge models. Specifically, the performance of LMMs should be evaluated using different versions of judge models, such as Claude-3-Opus, GPT-4o-20240806, GPT-4o-20240513, GPT-4v-20240409, GPT-4-1106-preview, or even open-sourced models (e.g., Qwen2.5 and LLaMA-3.1). This comprehensive approach will help in identifying any model-specific biases and in providing a more reliable assessment of the LMMs.

**Questions:**

I do not have any further questions. Please refer to the "weaknesses" section.

---

> ### Author Response · Authors · 2024-11-21
> **Response to Reviewer yNxN**
>
> Thank you for your valuable feedback on our paper. We appreciate your insights into the importance of robust evaluation metrics and variability considerations for MIA-Bench, and we are happy to address your concerns below.
>
> > Q1: Standard deviation reporting
>
> A1: In the table below we show STD information of the evaluation scores. For each model, we run inference for three times on MIA-Bench and compute STD of total score.
>
> | Model           | Fuyu-8b   | InstructBLIP-13b | Kosmos-2  | mPLUG-Owl2  | InternVL-Chat-v1.5 | Sphinx | Qwen-VL-Chat | LLaVA-1.5-7b | LLaVA-1.5-13b | LLaVA-1.6-7b | LLaVA-1.6-13b | LLaVA-1.6-34b | Idefics-2-8b | Gemini-1.0-Pro | Claude-3-Opus | Claude-3-Haiku | Claude-3-Sonnet | GPT-4v |
> |------------------|-----------|------------------|-----------|-------------|--------------------|--------|--------------|--------------|---------------|--------------|---------------|---------------|--------------|----------------|---------------|----------------|----------------|--------|
> | STD             | 0.00747   | 0.00586          | 0.00808   | 0.00096     | 0.01209            | 0.00517| 0.00373      | 0.01015      | 0.01134       | 0.0106       | 0.00273       | 0.01295       | 0.01493      | 0.00702        | 0.01724       | 0.00947        | 0.00465         | 0.00326 |
>
>
> > Q2: Detailed generation configuration
>
> A2: The setting of GPT-4o is set to default. Top-p and temperature default to 1. More settings can be found here in their official reference: https://platform.openai.com/docs/api-reference/chat
>
> > Q3: Impact of generation configuration
>
> A3: The common practice of using GPT-4/GPT-4v/GPT-4o as a judge is to use default settings.
>
> >Q4:  Performance across different judge models
>
> A4: In an attempt to analyze if GPT-4o’s evaluation is reliable, we used Claude-3-opus as a second judge and reported results in section ‘Other external models as the judge‘, from line 417 to line 424.
>
> We hope that our response addressed all your concerns. Thank you again for your feedback.

---

> > ### Comment · Reviewer_yNxN · 2024-11-22
> > **Post Rebuttal Comments by Reviewer yNxN**
> >
> > I believe the authors have not catched my key points, and this response is perfunctory. Specifically:
> >
> > 1. Towards Q1, more details should be reported regarding the STD.
> >       - Does the STD result from *multiple inferences using the same model*, or from *evaluating the same inference results using the same judge model?*
> >       - Why is the STD so small? For example, is the STD for Fuyu-8B 0.00747 (0.747%) or 0.00747%? The main table reports 24.52% for Fuyu-8B if my understanding is correct, so clarity on the STD value is essential.
> >
> > 2. Towards Q3, it is highly recommended to conduct multiple times using *the same judge model under the same inference results* with *different generation configurations of the judge model*. This will help ensure the robustness and reliability of the evaluation results.
> >
> > 3. Towards Q4, the evaluation should incorporate more judge models rather than relying solely on Claude-3-opus. Using a variety of judge models will provide a more comprehensive and balanced assessment of the performance.
> >
> > If the authors continue to provide perfunctory responses and do not take my concerns seriously, I will consider lowering my score.

---

> > > ### Author Response · Authors · 2024-11-24
> > > **Response to Reviewer yNxN - Part One**
> > >
> > > Thank you for your additional comments. Here we answer your questions one by one:
> > >
> > > > Q1 Does the STD result from multiple inferences using the same model, or from evaluating the same inference results using the same judge model?
> > >
> > > A1: We ran inference for three times on MIA-Bench, then evaluated the three different inference results using GPT-4o.
> > >
> > > > Q2 Why is the STD so small? For example, is the STD for Fuyu-8B 0.00747 (0.747%) or 0.00747%? The main table reports 24.52% for Fuyu-8B if my understanding is correct, so clarity on the STD value is essential.
> > >
> > > A2: 0.00747 means 0.747%. Below we update the table. The evaluation scores from multiple runs are pretty consistent, thus the STD is small.
> > >
> > > | Model | Fuyu-8b | InstructBLIP-13b | Kosmos-2 | mPLUG-Owl2 | InternVL-Chat-v1.5 | Sphinx | Qwen-VL-Chat | LLaVA-1.5-7b | LLaVA-1.5-13b | LLaVA-1.6-7b | LLaVA-1.6-13b | LLaVA-1.6-34b | Idefics-2-8b | Gemini-1.0-Pro | Claude-3-Opus | Claude-3-Haiku | Claude-3-Sonnet | GPT-4v |
> > > |------------------|-----------|------------------|-----------|-------------|--------------------|--------|--------------|--------------|---------------|--------------|---------------|---------------|--------------|----------------|---------------|----------------|----------------|--------|
> > > | STD | 0.747% | 0.586% | 0.808% | 0.096% | 1.209% | 0.517%| 0.373% | 1.015% | 1.134% | 1.06% | 0.273% | 1.295% | 1.493% | 0.702% | 1.724% | 0.947% | 0.465% | 0.326% |

---

> > > ### Author Response · Authors · 2024-11-24
> > > **Response to Reviewer yNxN - Part Two**
> > >
> > > > Q3 Towards Q3, it is highly recommended to conduct multiple times using the same judge model under the same inference results with different generation configurations of the judge model. This will help ensure the robustness and reliability of the evaluation results.
> > >
> > > A3: We set temperature to default (0) as this provides a more deterministic result; higher temperature increases the randomness which is not what we want from the judge. Top-p controls the diversity of generated text, this is by common practice of MLLM benchmarks to set to default as well, as the judge's main purpose is generating evaluation scores. Following your suggestion, we conducted additional evaluations using the gpt-4o-2024-11-20 model under different temperature settings (0, 0.1, and 0.2) and top p (0.8, 0.9, 1) to assess the robustness and reliability of the evaluation results.  Below are the result tables.
> > >
> > > | Model | Score by gpt-4o-2024-11-20 (temp=0) | Score by gpt-4o-2024-11-20 (temp=0.1) | Score by gpt-4o-2024-11-20 (temp=0.2) |
> > > |---------------------|-------------------------------------|--------------------------------------|--------------------------------------|
> > > | GPT-4o | 89.94 | 91.04 | 90.73 |
> > > | Claude-3-Opus | 84.89 | 87.66 | 87.41 |
> > > | Reka | 82.68 | 84.13 | 83.30 |
> > > | MiniCPM-Llama3-V-2.5| 78.75 | 79.18 | 79.55 |
> > > | Gemini | 76.32 | 76.20 | 76.76 |
> > > | LLaVA-1.5-13b | 66.05 | 67.63 | 67.25 |
> > > | ShareGPT4v | 65.72 | 66.97 | 67.18 |
> > > | Idefics-2-8b | 53.61 | 53.66 | 53.92 |
> > >
> > > Table 1: Evaluation score by gpt-4o-2024-11-20 with different temperature.
> > >
> > > | Model | Score by gpt-4o-2024-11-20 (top p=1) | Score by gpt-4o-2024-11-20 (top p=0.9) | Score by gpt-4o-2024-11-20 (top p=0.8) |
> > > |---------------------|--------------------------------------|---------------------------------------|---------------------------------------|
> > > | GPT-4o | 89.94 | 89.55 | 90.80 |
> > > | Claude-3-Opus | 84.89 | 86.81 | 86.68 |
> > > | Reka | 82.68 | 83.48 | 83.60 |
> > > | MiniCPM-Llama3-V-2.5| 78.75 | 78.95 | 79.35 |
> > > | Gemini | 76.32 | 75.73 | 75.69 |
> > > | LLaVA-1.5-13b | 66.05 | 67.26 | 67.43 |
> > > | ShareGPT4v | 65.72 | 67.18 | 66.88 |
> > > | Idefics-2-8b | 53.61 | 53.74 | 54.29 |
> > >
> > > Table 2: Evaluation score by gpt-4o-2024-11-20 with different top p.
> > >
> > > The results demonstrate consistent ranking across models, with minimal fluctuations in the scores despite changes in temperature. This consistency indicates that the judge model’s scoring is stable across varying generation configurations, further validating the robustness of our evaluation framework. The minor score variations observed are within an acceptable range.

---

> > > ### Author Response · Authors · 2024-11-24
> > > **Response to Reviewer yNxN - Part Four**
> > >
> > > | Model | Total Score | Description | Length Limit | Genres | Grammar | Mention | Math | Perspective | OCR |
> > > |---------------------|-------------|-------------|--------------|----------|----------|----------|----------|-------------|----------|
> > > | GPT-4o | 0.909704 | 0.927875 | 0.912371 | 0.942057 | 0.862434 | 0.900441 | 0.857143 | 0.916667 | 0.882883 |
> > > | Claude-3-Opus | 0.856077 | 0.890721 | 0.868490 | 0.917070 | 0.774590 | 0.819386 | 0.861111 | 0.725000 | 0.809524 |
> > > | Reka | 0.839905 | 0.894752 | 0.785088 | 0.905643 | 0.713661 | 0.801667 | 0.925926 | 0.657407 | 0.828571 |
> > > | MiniCPM-Llama3-V2.5| 0.798023 | 0.828916 | 0.771795 | 0.823087 | 0.751944 | 0.763976 | 0.721264 | 0.816667 | 0.841880 |
> > > | Gemini-1.0-Pro | 0.773569 | 0.817422 | 0.735470 | 0.788911 | 0.797814 | 0.683020 | 0.866071 | 0.870370 | 0.806373 |
> > > | LLaVA-1.5-7b | 0.683947 | 0.758817 | 0.703750 | 0.674046 | 0.630208 | 0.617620 | 0.425287 | 0.800000 | 0.602564 |
> > > | ShareGPT4v | 0.689046 | 0.800461 | 0.657738 | 0.608733 | 0.654762 | 0.601754 | 0.500000 | 0.800000 | 0.743056 |
> > > | Idefics-2-8b | 0.541755 | 0.560243 | 0.619318 | 0.489276 | 0.646825 | 0.455342 | 0.405556 | 0.375000 | 0.627193 |
> > >
> > > **Table 5**: Details of model scores evaluated by gpt-4o-2024-05-13.
> > >
> > > | Model | Total Score | Description | Length Limit | Genres | Grammar | Mention | Math | Perspective | OCR |
> > > |---------------------|-------------|-------------|--------------|----------|----------|----------|----------|-------------|----------|
> > > | GPT-4o | 0.899410 | 0.909379 | 0.916204 | 0.969395 | 0.854885 | 0.861247 | 0.920290 | 0.907407 | 0.878378 |
> > > | Claude-3-Opus | 0.848949 | 0.861543 | 0.871686 | 0.896552 | 0.797170 | 0.808777 | 0.846154 | 0.645833 | 0.865741 |
> > > | Reka | 0.826844 | 0.881841 | 0.809259 | 0.873276 | 0.725390 | 0.770225 | 0.814103 | 0.750000 | 0.819444 |
> > > | MiniCPM-Llama3-V2.5| 0.787537 | 0.818813 | 0.790246 | 0.795796 | 0.768182 | 0.736359 | 0.676667 | 0.716667 | 0.828125 |
> > > | Gemini-1.0-Pro | 0.763240 | 0.814379 | 0.750000 | 0.785159 | 0.757682 | 0.672255 | 0.758333 | 0.785714 | 0.776042 |
> > > | LLaVA-1.5-7b | 0.660472 | 0.751873 | 0.661822 | 0.649851 | 0.498512 | 0.572719 | 0.516667 | 0.750000 | 0.571429 |
> > > | ShareGPT4v | 0.657186 | 0.765309 | 0.632682 | 0.557578 | 0.583333 | 0.545104 | 0.464286 | 0.675000 | 0.717742 |
> > > | Idefics-2-8b | 0.536134 | 0.589964 | 0.541887 | 0.455882 | 0.611582 | 0.449821 | 0.406667 | 0.527778 | 0.576190 |
> > >
> > > **Table 6**: Details of model scores evaluated by gpt-4o-2024-11-20.
> > >
> > > Due to legal agreements the authors are obligated to adhere to, we have been requested not to use other close sourced MLLMs (eg, Clause-3-Opus, or Reka) for full rounds of evaluation. We did not use GPT-4v series because they are deprecated, and we did not use gpt-4-1106-preview, LLaMA-3.1, and Qwen2.5 as they are LLMs, but our judge needs to be a vision-language model to take in images as images are mandatory in the evaluation process. We chose not to convert the images into textual captions then use LLMs to evaluate because 1) our instructions a lot of times require MLLMs to focus on details in the images; if images are converted into captions, these details can be lost, thus the evaluation will be inaccurate, 2) MLLMs like GPT-4o and GPT-4v can serve as judges better than LLMs. As for open-source models; in our evaluation result table, we showed that their performance on MIA-Bench are not as good as proprietary models, thus not suitable as judges for this task.
> > >
> > > We hope that we addressed your concerns. Thank you again for your review.

---

> > > > ### Comment · Reviewer_yNxN · 2024-11-25
> > > > **Reply to New Results**
> > > >
> > > > I really appreciate the authors' sufficient empirical results, where I found that different generation configurations and judge models contribute to similar rankings.
> > > >
> > > > Therefore, I will keep my rating and recommend acceptance.

---

> > ### Author Response · Authors · 2024-11-24
> > **Response to Reviewer yNxN - Part Three**
> >
> > > Q4 Towards Q4, the evaluation should incorporate more judge models rather than relying solely on Claude-3-opus. Using a variety of judge models will provide a more comprehensive and balanced assessment of the performance.
> >
> > A4: The judge we use for MIA-Bench is GPT-4o. To alleviate this concern that GPT-4o may favorably score its own responses, as reported in the paper, we use Claude-3-opus, a strong performer, to evaluate responses from GPT-4o and itself, and compare their scores with each other to double check if GPT-4o is the best performing model on this benchmark. We find that even using Claude-3 Opus to score its own and GPT-4o's responses, GPT-4o still achieves a superior score. Based on this observation, we use GPT-4o for evaluation by default, to ensure the correctness of evaluation. It's common practice to use one judge model instead of multiple to evaluate inference result for efficiency. Here we list a few benchmarks that use one version of GPT as the judge: LLaVA-Bench[1] (NeurIPS 2023), MMBench[2] (ECCV 2024), MathVista[3] (ICLR 2024), HallusionBench[4] (CVPR 2024), CV-Bench[5] (NeurIPS 2024),  MM-Vet[6], MMHAL-BENCH[7], MLVU[8], etc.
> >
> > We conducted additional experiments following your suggestion to use different judges, namely gpt-4o-2024-11-20, gpt-4o-2024-05-13, chatgpt-4o-latest, and gpt-4o-mini-2024-07-18. The results are updated in the appendix: A.4 Comparison of Scores and Rankings across Different Judge Models. We also paste the result tables here. We evaluated eight MLLMs using these four judges. The ranking is consistent with a minor difference. (LLaVA-1.5-13b and ShareGPT4, when evaluated by gpt-4o-2024-05-13, has a different ranking order from when evaluated by other judges. This is not surprising though, as their performance is similar.)
> >
> > | Model | Score by chatgpt-4o-latest | Ranking by chatgpt-4o-latest | Score by gpt-4o-2024-11-20 | Ranking by gpt-4o-2024-11-20 | Score by gpt-4o-2024-05-13 | Ranking by gpt-4o-2024-05-13 | Score by gpt-4o-mini-2024-07-18 | Ranking by gpt-4o-mini-2024-07-18 |
> > |---------------------|----------------------------|------------------------------|----------------------------|------------------------------|----------------------------|------------------------------|----------------------------------|----------------------------------|
> > | GPT-4o | 89.69 | 1 | 89.94 | 1 | 90.97 | 1 | 81.36 | 1 |
> > | Claude-3-Opus | 86.16 | 2 | 84.89 | 2 | 85.61 | 2 | 78.95 | 2 |
> > | Reka | 83.09 | 3 | 82.68 | 3 | 83.99 | 3 | 77.70 | 3 |
> > | MiniCPM-Llama3-V2.5| 78.10 | 4 | 78.75 | 4 | 79.80 | 4 | 73.72 | 4 |
> > | Gemini | 75.77 | 5 | 76.32 | 5 | 77.36 | 5 | 67.45 | 5 |
> > | LLaVA-1.5-13b | 66.78 | 6 | 66.05 | 6 | 68.39 | 7 | 61.54 | 6 |
> > | ShareGPT4v | 66.61 | 7 | 65.72 | 7 | 68.90 | 6 | 60.30 | 7 |
> > | Idefics-2-8b | 53.51 | 8 | 53.61 | 8 | 54.18 | 8 | 44.28 | 8 |
> >
> > **Table 3**: Comparison of scores and rankings across different judge models.
> >
> > | Model | Total Score | Description | Length Limit | Genres | Grammar | Mention | Math | Perspective | OCR |
> > |---------------------|-------------|-------------|--------------|----------|----------|----------|----------|-------------|----------|
> > | GPT-4o | 0.896893 | 0.906288 | 0.917996 | 0.955952 | 0.830508 | 0.867949 | 0.846667 | 0.833333 | 0.896396 |
> > | Claude-3-Opus | 0.861628 | 0.895363 | 0.866039 | 0.927730 | 0.807018 | 0.820549 | 0.857639 | 0.666667 | 0.800926 |
> > | Reka | 0.830885 | 0.869867 | 0.821685 | 0.883403 | 0.795597 | 0.772894 | 0.813218 | 0.675000 | 0.848485 |
> > | MiniCPM-Llama3-V2.5| 0.780966 | 0.831197 | 0.766026 | 0.796257 | 0.726190 | 0.722037 | 0.691358 | 0.656250 | 0.768018 |
> > | Gemini-1.0-Pro | 0.757733 | 0.793860 | 0.724138 | 0.745455 | 0.854167 | 0.670349 | 0.816056 | 0.822917 | 0.822581 |
> > | LLaVA-1.5-7b | 0.667826 | 0.743137 | 0.638889 | 0.675827 | 0.571212 | 0.594505 | 0.500000 | 0.758333 | 0.596774 |
> > | ShareGPT4v | 0.666092 | 0.773905 | 0.661290 | 0.573904 | 0.562500 | 0.570722 | 0.458333 | 0.638889 | 0.695238 |
> > | Idefics-2-8b | 0.535057 | 0.597963 | 0.531810 | 0.483768 | 0.593056 | 0.452361 | 0.326087 | 0.458333 | 0.569444 |
> >
> > **Table 4**: Details of model scores evaluated by chatgpt-4o-latest.

---

### Official Review · Reviewer_drhM · 2024-11-09

**Soundness:** 3
**Presentation:** 3
**Contribution:** 3
**Rating:** 6
**Confidence:** 4

**Summary:**

This paper introduces MIA-Bench, a benchmark designed to evaluate multimodal large language models (MLLMs) on strict adherence to complex instructions. Based on a dataset of 400 image-instruction pairs, MIA-Bench assesses the performance of 29 popular MLLMs, highlighting that current MLLMs struggle with precise instruction adherence. In addition, the paper also explored the supervised fine-tuning (SFT) method based on the LLaVA-NeXT model and achieved positive results, demonstrating the potential effectiveness of this method in improving model instruction adherence.

**Strengths:**

(1) MIA-Bench fills a gap in existing benchmarks by focusing on the ability of multimodal large language models (MLLMs) to adhere to complex instructions, a crucial yet previously underexplored area of evaluation. This benchmark’s design reveals potential deficiencies in models when following strict, multi-layered instructions, supporting the practical deployment of multimodal models in complex, instruction-based tasks.
(2) The MIA-Bench dataset consists of 400 image-instruction pairs covering various instruction types, such as description, length limit, genre, grammar, and OCR. With data from diverse sources, it reflects the variety found in real-world scenarios. This diverse set of instructions enhances the benchmark’s comprehensiveness and real-world applicability.
(3) The paper provides a systematic evaluation of 29 popular MLLMs, analyzing their performance across different instruction categories. This large-scale comparison offers researchers a detailed performance reference and targeted insights for future model improvements.

**Weaknesses:**

(1) Although the SFT experiments in the paper demonstrate improved performance in adhering to complex instructions, they may lack in terms of model generalization. The results may be specific to the LLaVA-NeXT model, with no validation of applicability to other models, leaving it unproven whether SFT is equally effective for enhancing complex instruction adherence in MLLMs of different architectures and sizes.
(2) The differences between MIA-Bench and other benchmarks are indeed striking, and the authors believe that this may indicate that MIA-Bench's unique design focuses more on evaluating the model's strict instruction-following ability. However, the authors' explanation does not completely rule out the possibility that MIA-Bench itself may have design biases. This explanation is based on speculative experimental results and has not been verified in depth by sufficient experiments.
(3) Although the authors introduced Claude-3-Opus for comparison to mitigate scoring bias, the two models may exhibit similar biases in evaluation. Therefore, using Claude-3-Opus as the sole comparison tool may be insufficient to fully reveal potential scoring biases in GPT-4o.

**Questions:**

please refer to the weakness part.

---

> ### Author Response · Authors · 2024-11-25
> **Response to Reviewer drhM - Part Two**
>
> > Q3. Although the authors introduced Claude-3-Opus for comparison to mitigate scoring bias, the two models may exhibit similar biases in evaluation. Therefore, using Claude-3-Opus as the sole comparison tool may be insufficient to fully reveal potential scoring biases in GPT-4o.
>
> A3. In our evaluation, we use GPT-4o as the default judge model because of two reasons: The most widely recognized free-form evaluation benchmarks currently adopt ChatGPT-series models as their judge, as they represent the state-of-the-art MLLMs available. Examples of such benchmarks include LLaVA-Bench [1] (NeurIPS 2023), MMBench [2] (ECCV 2024), MathVista [3] (ICLR 2024), HallusionBench [4] (CVPR 2024), and CV-Bench [5] (NeurIPS 2024), etc. To align with this common practice, we have chosen GPT-4o as the default evaluation model.
>
> To further assess the reliability of the scoring, as outlined in Section 3.2, we employ the second-highest-performing model on MIA-Bench, Claude-3 Opus, to evaluate its own responses alongside those of GPT-4o. Interestingly, Claude-3 Opus also favors the GPT-4o responses over its own on MIA-Bench. This preliminary experiment demonstrates consistency across these two judge models.
>
> We sincerely appreciate your constructive feedback, which will help improve the quality and depth of our work. Thank you again for your time and valuable insights.
>
> References:
> [1] Haotian Liu, Chunyuan Li, Qingyang Wu, and Yong Jae Lee. Visual instruction tuning, NeurIPS 2023.
> [2] Yuan Liu, Haodong Duan, Yuanhan Zhang, Bo Li, Songyang Zhang, Wangbo Zhao, Yike Yuan, Jiaqi Wang, Conghui He, Ziwei Liu, Kai Chen, and Dahua Lin. Mmbench: Is your multi-modal model an all-around player?, ECCV 2024.
> [3] PanLu,HritikBansal,TonyXia,JiachengLiu,ChunyuanLi,HannanehHajishirzi,HaoCheng,Kai-Wei Chang, Michel Galley, and Jianfeng Gao. Mathvista: Evaluating mathematical reasoning of foundation models in visual contexts, 2024.
> [4] Tianrui Guan, Fuxiao Liu, Xiyang Wu, Ruiqi Xian, Zongxia Li, Xiaoyu Liu, Xijun Wang, Lichang Chen, Furong Huang, Yaser Yacoob, Dinesh Manocha, and Tianyi Zhou. Hallusionbench: An advanced diagnostic suite for entangled language hallucination and visual illusion in large vision-language models, 2024.
> [5] Shengbang Tong, Ellis Brown, Penghao Wu, Sanghyun Woo, Manoj Middepogu, Sai Charitha Akula, Jihan Yang, Shusheng Yang, Adithya Iyer, Xichen Pan, Austin Wang, Rob Fergus, Yann LeCun, and Saining Xie. Cambrian-1: A fully open, vision-centric exploration of multimodal llms, 2024.

---

### Meta-Review · Area_Chair_gpDV · 2024-12-21

**Metareview:**

This paper introduces a new MIA-Bench benchmark specifically designed for MLLM's complex instruction following ability study. The authors' major contribution is a new dataset containing 400 image-instruction pairs, where the instructions are written by humans with high quality. Besides, the authors provided a comprehensive evaluation of the existing MLLMs and provided SFT positive results. The weaknesses and concerns are the model generalization issue, the MIA-Bench's design biases, the diverse model judgment issue, and the potential data leakage issue. After rebuttal, the authors addressed most of these concerns, and all reviewers agreed with the acceptance. Meanwhile, the value of the new benchmark with 400 high-quality image-instruction pairs is highlighted. Therefore, the final recommendation is accept.

**Additional Comments On Reviewer Discussion:**

This submission received four reviews.
Reviewer drhM's major concerns were about the SFT model generalization issue, the MIA-Bench's design biases issue, and the solo Claude-3-Opus judgment issue. The authors provided one-by-one responses to these questions with new experimental results. The reviewers did not provide final comments, but the initial score is a marginal acceptance.
Reviewer yNxN's concerns were mainly about the robustness of the evaluation. The authors provided comprehensive new results according to the reviewer's suggestions. The reviewer's final decision is a marginal acceptance.
Reviewer PupP's major concern was the potential data leakage issue. The authors acknowledged that images may be covered by previous models, but emphasized their instructions are of high quality and written by humans, especially composed of multiple levels of instructions. The reviewer accepted the explanation and improved the score from 5 to 6.
Reviewer UieD's concerns were about the value of complex instruction for MLLMs and the issue of performance. After rebuttal, the reviewer appreciated the contribution of the new dataset and improved the score from 5 to 6.
Considering all reviewers' opinions and the discussion, the final recommendation is accept.

---

### Decision · Program_Chairs · 2025-01-22

Accept (Poster)